# OmniConsistency: Learning Style-Agnostic Consistency from Paired Stylization Data

Yiren Song*   Cheng Liu*   Mike Zheng Shou†

Show Lab, National University of Singapore

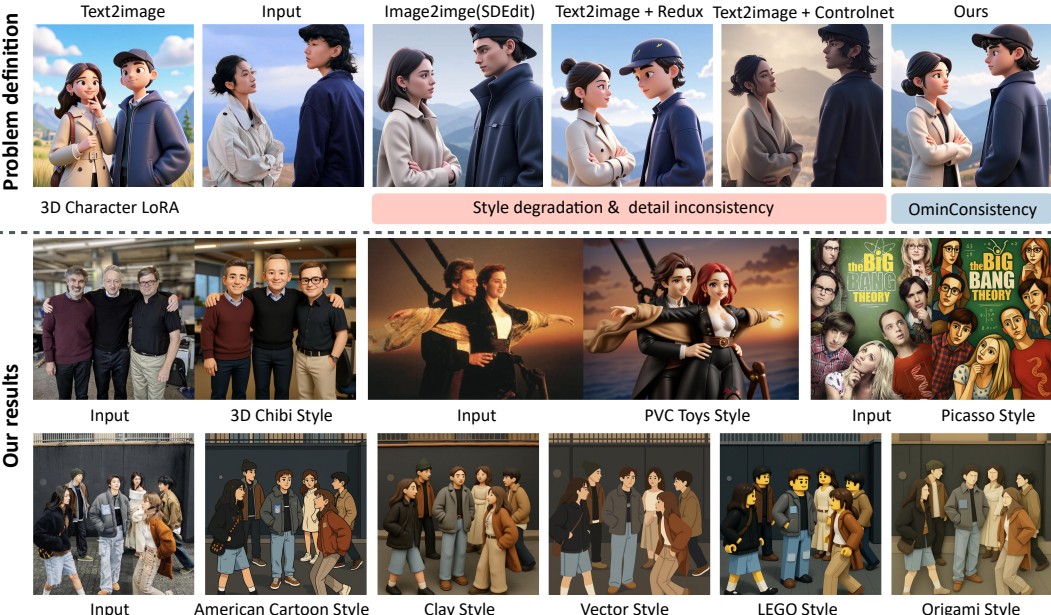

Figure 1: Our method achieves style-consistent and structure-preserving image stylization under diverse scenes and unseen style LoRAs, outperforming existing baselines without style degradation.

## Abstract

Diffusion models have advanced image stylization significantly, yet two core challenges persist: (1) maintaining consistent stylization in complex scenes, particularly identity, composition, and fine details, and (2) preventing style degradation in image-to-image pipelines with style LoRAs. GPT-4o's exceptional stylization consistency highlights the performance gap between open-source methods and proprietary models. To bridge this gap, we propose **OmniConsistency**, a universal consistency plugin leveraging large-scale Diffusion Transformers (DiTs). OmniConsistency contributes: (1) an in-context consistency learning framework trained on aligned image pairs for robust generalization; (2) a two-stage progressive learning strategy decoupling style learning from consistency preservation to mitigate style degradation; and (3) a fully plug-and-play design compatible with arbitrary style LoRAs under the Flux framework. Extensive experiments show that OmniConsistency significantly enhances visual coherence and aesthetic quality, achieving performance comparable to commercial state-of-the-art model GPT-4o. Code is released at https://github.com/showlab/OmniConsistency

---

*Equal contribution.

†Corresponding author.

39th Conference on Neural Information Processing Systems (NeurIPS 2025).

# 1  Introduction

Image stylization aims to transfer artistic styles to target images. With the emergence of diffusion models, the mainstream approach has shifted toward fine-tuning pretrained models via Low-Rank Adaptation (LoRA) [17], coupled with image-to-image (I2I) inference pipelines and consistency modules (e.g., ControlNet [49]), significantly enhancing stylization quality. Recently, open-source communities have released numerous stylization-oriented LoRA modules. Additionally, methods like InstantStyle [43] and IPAdapter [48] enable tuning-free stylization via adapter modules pretrained on large-scale datasets, allowing efficient style transfer without task-specific fine-tuning.

Despite recent progress, current image stylization methods face three key challenges: (1) Limited consistency between stylized outputs and inputs—existing modules (e.g., ControlNet) ensure global alignment but fail to preserve fine semantics and details in complex scenes. (2) Style degradation in image-to-image (I2I) settings—LoRA and IPAdapter often yield lower style fidelity than in text-to-image generation, as Figure. 1 shown. (3) Lack of flexibility in layout control—methods relying on rigid conditions (e.g., edges, sketches, poses) struggle to support creative structure changes like chibi-style transformation.

These issues significantly restrict the practical performance of existing methods, motivating this research. To address these challenges, we propose OmniConsistency, a general consistency plugin based on the Diffusion Transformer architecture, combined with an in-context learning strategy, specifically designed for image stylization tasks. OmniConsistency precisely preserves image semantics and details during style transfer in a style-agnostic manner.

To effectively support model training, we meticulously constructed a high-quality, multi-source stylization dataset, covering 22 different styles and totaling 2,600 image pairs. Data sources include manually drawn illustrations and GPT-4o-guided [1] generation of highly consistent stylized images. After rigorous manual selection, we obtained a reliable paired dataset suitable for consistency model training.

To decouple style learning from consistency learning, we propose a two-stage decoupled training framework along with a rolling LoRA Bank Loader mechanism: In the first stage, we independently train LoRA models on style-specific data to build a LoRA Bank; in the second stage, we attach the pretrained style LoRA modules onto a Diffusion Transformer [29] backbone and train the consistency module using corresponding image pairs (original and stylized images). The second-stage training explicitly targets structural and semantic consistency, preventing the consistency module from absorbing any specific style features. To ensure style-agnostic capability, the LoRA modules and their corresponding data subsets are periodically switched during training iterations, ensuring stable consistency performance across diverse styles and achieving strong generalization, supporting plug-and-play integration with arbitrary style LoRA modules.

Furthermore, to achieve more flexible layout control, we forego traditional explicit geometric constraints (such as edges, sketches, poses) commonly used in previous methods. Instead, we adopt a more flexible implicit control strategy, utilizing only the original image itself as the conditioning input. This approach allows OmniConsistency to better balance style expression and structural consistency, especially suitable for tasks involving significant character proportion transformations, such as chibi-style generation. Through a data-driven approach, the model autonomously learns composition and semantic consistency mappings from paired data, further enhancing its generalization capabilities.

In summary, our key contributions are as follows:

1. We propose OmniConsistency, a universal consistency plugin based on Diffusion Transformers with in-context learning, significantly enhancing visual consistency in I2I stylization tasks in a style-agnostic manner.

2. We design a two-stage, style-consistency disentangled training strategy and innovatively introduce a rolling LoRA Bank loader mechanism, substantially improving consistency generalization across diverse styles. Moreover, we propose a lightweight Consistency LoRA Module and a Conditional Token Mapping scheme, effectively improving computational efficiency.

3. We build and release a diverse stylization dataset and benchmark for image stylization consistency and introduce a standardized evaluation protocol based on GPT-4o, facilitating comprehensive performance assessments.

## 2 Related Works

### 2.1 Diffusion Models

Image generation has experienced a major paradigm shift in recent years, with diffusion models [16] increasingly surpassing GANs [10] as the dominant approach, thanks to their superior image quality and training stability. Diffusion models are widely applied in areas such as image synthesis [32**?** ], image editing [3, 13, 51, 53, 47, 18, 11, 19, 9], video gneration [12, 2, 5, 42], and process generation [38, 39, 37]. Early successes in this field primarily relied on U-Net-based denoising architectures. Representative works include Stable Diffusion (SD) [32], its improved variant SDXL [30], and several other foundational models, all of which demonstrated the strong potential of diffusion models for high-fidelity image synthesis. More recently, the field has evolved toward transformer-based architectures, most notably through the emergence of the Diffusion Transformer (DiT) framework. State-of-the-art models such as SD3 [6], FLUX [21], and HunyuanDiT [23] leverage the scalability and representation power of transformers to push generation quality even further. Compared with their U-Net-based predecessors, DiT models exhibit markedly better output fidelity and prompt alignment, setting a new standard for diffusion-based generation.

### 2.2 Stylized Image Generation

Recent diffusion-based methods have enabled efficient style transfer via tuning-free adapters such as IP-Adapter [48], Style-Adapter [44], and StyleAlign [45]. These approaches extract style embeddings from a single reference image and inject them into the generation process using cross-attention layers. However, many visual styles cannot be fully captured by a single image. For instance, the Ghibli aesthetic involves consistent design across characters, environments, and objects. In practice, training style-specific LoRA modules on multiple examples remains the most effective and widely adopted approach [33, 20, 4, 34, 7], offering stronger generalization and stylization quality in text-to-image generation. Yet, when these LoRA modules are applied to image-to-image translation or editing tasks, they often suffer from style degradation due to structural constraints imposed by modules like ControlNet [49]. This results in diminished style expressiveness and visual inconsistency. To resolve this, we propose OmniConsistency, a plug-and-play consistency module that enhances style retention under structural guidance. Rather than replacing LoRA, our method augments it, ensuring faithful style preservation even in controlled editing scenarios.

### 2.3 Condition-guided Diffusion Models

Conditional diffusion models have rapidly evolved, with increasingly refined mechanisms for controllable image generation. Broadly, conditioning signals fall into two categories: semantic conditions, which guide high-level content (e.g., reference images of subjects or objects), and spatial conditions, which constrain structural layout (e.g., edge maps, depth cues, or human poses). Earlier approaches, typically built on U-Net backbones, adopted two main paradigms: attention-based modules such as IP-Adapter [48] and SSR-Encoder [50] focused on integrating semantic information, while residual-based methods like ControlNet [49] and T2I-Adapter [27] were designed to maintain spatial fidelity. With the emergence of transformer-based diffusion architectures (e.g., DiT [29]), conditioning strategies have shifted toward more unified and efficient token-based designs. Recent methods like OminiControl [40] and EasyControl [52] treat both semantic and spatial conditions as token sequences, enabling seamless integration with transformer blocks, and inspired subsequent approaches [36, 25, 35]. This transition simplifies the overall design, improves scalability, and facilitates more effective handling of multimodal inputs. The shift from U-Net to DiT-based conditioning reflects a broader trend in generative modeling: moving toward more modular, generalizable, and computation-efficient frameworks for controlled generation.

## 3 Methods

In Sec. 3.1, we introduce the overall architecture of our proposed method; in Sec. 3.2, we present the decoupled training strategy for style-consistency learning; in Sec. 3.3, we describe the consistency LoRA Module; in Sec. 3.4, we detail the position encoding interpolation; and in Sec. 3.5, we explain the composition and collection process of the paired dataset.

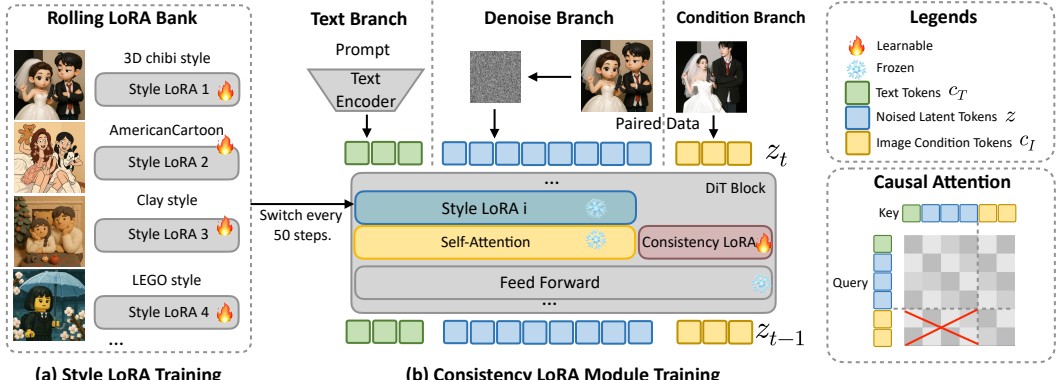

Figure 2: Illustration of OmniConsistency, consisting of style learning and consistency learning phases. (a) In the style learning phase, individual LoRA modules are trained on dedicated datasets to capture unique stylistic details. (b) The subsequent consistency learning phase optimizes consistency LoRA for structural and detail coherence across diverse stylizations, integrating pre-trained style LoRA dynamically.

## 3.1 Overall Architecture

The OmniConsistency framework is designed to achieve robust style-agnostic consistency in image stylization. As shown in Fig. 2, the method is composed of two coordinated components: a two-stage training pipeline and several plug-and-play architectural modules that enhance controllability and generalization.

In the training pipeline, we first build a style LoRA bank by independently fine-tuning LoRA modules for 22 styles. In the second stage, we train a consistency control module, referred to as consistency LoRA, on the same paired data while dynamically switching the style LoRA module in alignment with the training instance. This strategy decouples stylization from consistency and improves generalization across styles.

Beyond the training design, our framework introduces two architectural components to enhance achieve style-consistency disentanglement and efficien consistency control: (1) A **Consistency LoRA Module**, which injects condition-specific information through a dedicated low-rank adaptation path applied only to conditional branches; (2) A **Position-Aware Interpolation** and **Feature Reuse** enables the use of low-resolution condition images to guide high-resolution generation while strictly preserving spatial alignment. This design improves both training and inference efficiency. Together, these designs allow OmniConsistency to preserve semantic structure and fine details across diverse stylizations, while supporting flexible control and efficient computation.

## 3.2 Style-Consistency Decoupled Training

To address the limitations described above and further enhance OmniConsistency's robustness and flexibility, we introduce a novel two-stage decoupled training strategy that explicitly separates style learning from consistency preservation. This method contrasts conventional joint-training approaches that simultaneously optimize both style and consistency components, potentially causing conflicts and suboptimal convergence.

**Stage 1: Style Learning.** In this initial phase, we independently train multiple style-specific LoRA modules on dedicated datasets, each corresponding to one particular style (e.g., anime, oil painting, photorealism). These datasets consist of paired stylized images and their original counterparts. During training, each LoRA module is fine-tuned from the pretrained Diffusion Transformer backbone with a fixed learning rate of $1 \times 10^{-3}$ for 6,000 iterations. The primary objective at this stage is to accurately capture distinctive artistic elements, textures, color palettes, and stylistic details associated uniquely with each style. By isolating this process, we prevent interference from structural consistency constraints and create a style LoRA bank.

**Stage 2: Consistency Learning.** In the subsequent stage, we aim to learn a style-agnostic consistency module that can effectively preserve structural, semantic, and detailed consistency regardless of the

applied style. Specifically, we introduce a lightweight Consistency LoRA Module, which integrates seamlessly with pretrained style LoRA modules. During this phase, style LoRA modules from the first stage are dynamically loaded in a **Rolling LoRA Bank**, periodically switching between different style LoRAs along with their corresponding paired datasets during training iterations. This approach ensures the consistency module optimizes exclusively for preserving input content integrity, actively avoiding the absorption of specific stylistic traits.

Through this explicit decoupling of style and consistency training objectives and the introduction of novel techniques such as the rolling LoRA bank loader, our approach ensures both superior stylization quality and robust content preservation across diverse stylistic transformations.

### 3.3 Consistency LoRA Module

**LoRA Design for Consistency.** To efficiently incorporate conditional signals while preserving the stylization capacity of the diffusion backbone, we extend the FLUX [21] architecture with a dedicated consistency LoRA module applied only to the condition branch.

Conventional methods apply control modules to the main network layers [40], which disrupt style representation. In contrast, our design isolates consistency learning from the stylization pathway to ensure compatibility. Specifically, we leave the LoRA attachment points on the main diffusion transformer unoccupied, allowing arbitrary style LoRAs to be mounted independently. This branch-isolated design ensures compatibility between consistency learning and stylization, enabling both modules to operate without conflict or parameter entanglement.

Formally, given input features $Z_t, Z_n, Z_c$ for the text, noise, and condition branches, we define the standard QKV projections as:

$$Q_i = W_Q Z_i, \quad K_i = W_K Z_i, \quad V_i = W_V Z_i, \quad i \in \{t, n, c\} \tag{1}$$

where $W_Q, W_K, W_V \in \mathbb{R}^{d \times d}$ are shared projection matrices across branches. To inject conditional information more effectively, we apply LoRA transformations solely to the condition branch:

$$\Delta Q_c = B_Q A_Q Z_c, \quad \Delta K_c = B_K A_K Z_c, \quad \Delta V_c = B_V A_V Z_c \tag{2}$$

where $A_Q, A_K, A_V \in \mathbb{R}^{r \times d}$ and $B_Q, B_K, B_V \in \mathbb{R}^{d \times r}$ are low-rank adaptation matrices with $r \ll d$. The updated QKV for the condition branch becomes:

$$Q'_c = Q_c + \Delta Q_c, \quad K'_c = K_c + \Delta K_c, \quad V'_c = V_c + \Delta V_c \tag{3}$$

Meanwhile, the text and noise branches remain unaltered:

$$Q'_i = Q_i, \quad K'_i = K_i, \quad V'_i = V_i, \quad i \in \{t, n\} \tag{4}$$

This design ensures that consistency-related adaptation is introduced in an isolated manner, without interfering with the backbone's stylization capacity or other conditioning paths.

**Causal Attention.** Unlike Flux and prior controllable generation methods, we replace the original bidirectional attention with causal attention, a setting that follows EasyControl [52]. As shown in Fig. 2, we design a structured attention mask where condition tokens can only attend to each other and are blocked from accessing noise/text tokens, while the main branch (noise and text tokens) follows standard causal attention and can attend to the condition tokens. This design offers two key advantages: (1) the main branch maintains clean causal modeling during inference, avoiding interference from condition tokens; and (2) no additional LoRA parameters are introduced to the noise/text branch, preserving all tunable capacity for style LoRA and preventing conflicts between stylization and consistency. By enforcing this read-only conditioning mechanism, we improve editing controllability while maintaining a clear separation between style and structure.

### 3.4 Designs for Efficient and Scalable Conditioning

To improve the computational efficiency of transformer-based diffusion models, we introduce two complementary techniques: (1) **Conditional Token Mapping** for low-resolution conditional guidance, and (2) **Feature Reuse** for eliminating redundant computation across denoising steps.

**Conditional Token Mapping (CTM).** Concatenating full-resolution condition tokens with denoising tokens leads to high memory usage and inference latency. To address this, we use a low-resolution

condition image to guide high-resolution generation, with spatial alignment ensured via CTM. Given original resolution $(M, N)$ and condition resolution $(H, W)$, we define scaling factors:

$$S_h = \frac{M}{H}, \quad S_w = \frac{N}{W} \tag{5}$$

Each token $(i, j)$ in the downsampled condition maps to position $(P_i, P_j)$ in the high-resolution grid:

$$P_i = i \cdot S_h, \quad P_j = j \cdot S_w \tag{6}$$

This mapping preserves pixel-level correspondence between condition and output features, enabling structurally coherent guidance under significant resolution mismatch.

**Feature Reuse.** During standard diffusion, condition tokens remain fixed across all denoising steps, while latent tokens evolve. To reduce repeated computation, we cache the intermediate features of condition tokens—specifically their key-value projections in attention and reuse them throughout inference [28, 41]. This optimization significantly lowers inference time and GPU memory without sacrificing generation quality.

### 3.5 Dataset Collection

We construct a high-quality paired dataset entirely through GPT-4o-driven generation [1]. Specifically, we leverage GPT-4o to synthesize stylized versions of input images across 22 diverse artistic styles, as well as generate corresponding descriptive text annotations for both source and stylized images.

The input images are collected from publicly available internet sources and carefully curated to ensure legal compliance. To ensure semantic and structural consistency, we apply a human-in-the-loop filtering pipeline. Annotators review each generated image pair and remove those with issues such as gender mismatches, incorrect age or skin tone, detail distortions, pose discrepancies, inconsistent styles, or misaligned layouts. This rigorous filtering process is applied to over 5,000 candidate pairs, from which we curate 80–150 high-quality pairs per style, resulting in a total of 2,600 verified image pairs.

To promote diversity, the input images for each style are mutually exclusive, with complex scenes such as multi-person portraits. The dataset spans a wide range of styles—including anime, sketch, chibi, pixel-art, watercolor, oil painting, and cyberpunk—and will be publicly released to support future research in stylization and consistency modeling.

## 4 Experiments

### 4.1 Experiments Details

**Set up.** We adopt Flux 1.0 dev [21] as the pre-trained model. The dataset resolution is $1024{\times}1024$, while condition images are downsampled to $512{\times}512$ to reduce memory and computation, with high-resolution control achieved via conditional token mapping. The training is conducted in two stages: the first stage fine-tunes the style LoRA for 6,000 steps on a single GPU, using a learning rate of $1 \times 10^{-4}$ and a batch size of 1. The second stage trains the consistency module from scratch for 9,000 steps on 4 GPUs, with a per-GPU batch size of 1 (total batch size = 4) and the same learning rate. In this stage, every 50 steps, a style LoRA and its corresponding data are loaded from the LoRA bank to encourage multi-style generalization.

**Benchmark.** To evaluate our method against baseline approaches, we propose a new image-to-image benchmark consisting of 100 images with complex visual compositions, including group portraits, animals, architectural scenes, and natural landscapes. For fair comparison, we selected 5 style LoRA models from the LibLibAI [24] website for stylization and quantitative evaluation. These styles were not included in the LoRA Bank used during training. The five styles are comic, oil painting, PVC toys, sketch, and vector style.

**Baseline Methods.** In this section, we introduce the baseline methods. The compared approaches include: 1. Flux image-to-image pipeline (based on SDEdit) [26]; 2. Flux image-to-image pipeline with Redux [22]; 3. Flux text-to-image pipeline with Redux; 4. Flux image-to-image pipeline with ControlNet [49]; 5. Flux text-to-image pipeline with ControlNet; 6. GPT-4o [1], the most advanced commercial image stylization API. For ControlNet baselines, canny and depth maps are jointly used for conditioning, with each modality weighted at 0.5 and early stopping applied at 0.5.

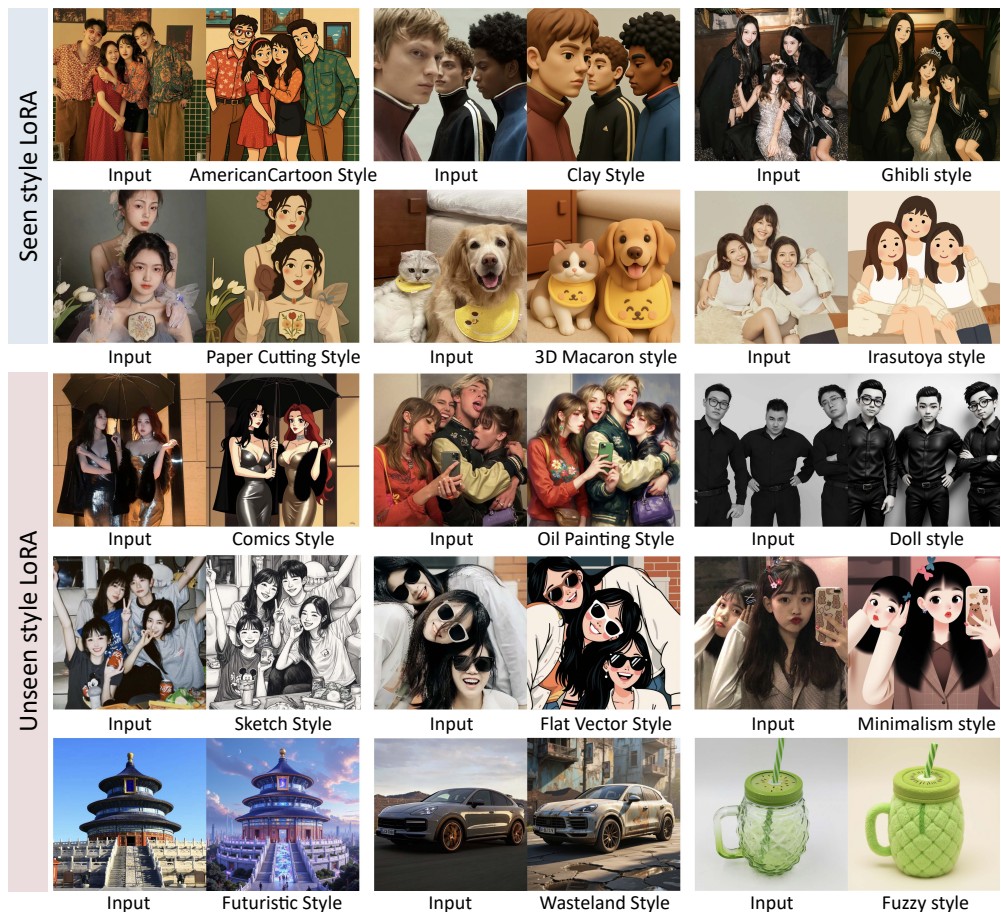

Figure 3: OmniConsistency can be combined with both seen and unseen style LoRA modules to achieve high-quality image stylization consistency, effectively preserving the semantics, structure, and fine details of the original image.

## 4.2 Evaluation Metrics

We evaluate our method from three aspects: **style consistency**, **content consistency**, and **text-image alignment**, using a benchmark of 100 test images with captions generated by GPT-4o. All image similarity metrics are computed using **DreamSim** [8], **CLIP Image Score** [31], and **GPT-4o Score**. For style consistency, we compare the stylized result with a reference generated by applying the same LoRA to the same prompt and seed. We also compute **FID** [15] and **CMMD** [46] over 1,000 samples (generated by repeating the benchmark 10 times with different seeds) to assess the impact of OmniConsistency on the style distribution. For content consistency, we measure similarity between the stylized image and the input image. For text-image alignment, we use the standard **CLIP Score** [14] to evaluate how well the output aligns with the input prompt.

## 4.3 Quantitative Evaluation

As shown in Table 1, our method achieves the best performance across five style consistency metrics and ranks among the top in content consistency. It also obtains the highest CLIP Score, indicating superior text-image alignment. These results demonstrate that our consistency-aware framework effectively balances stylization fidelity, semantic preservation, and prompt alignment. In terms of content consistency, Flux I2I + Redux achieves the highest CLIP Image Score; however, this advantage largely stems from its limited stylization strength and minimal visual transformation.

## 4.4 Qualitative Evaluation

As shown in Fig. 4, the T2I baseline reflects the expected stylization effect of the LoRA. The Redux method achieves reasonable stylization but suffers from poor content and structural consistency. The

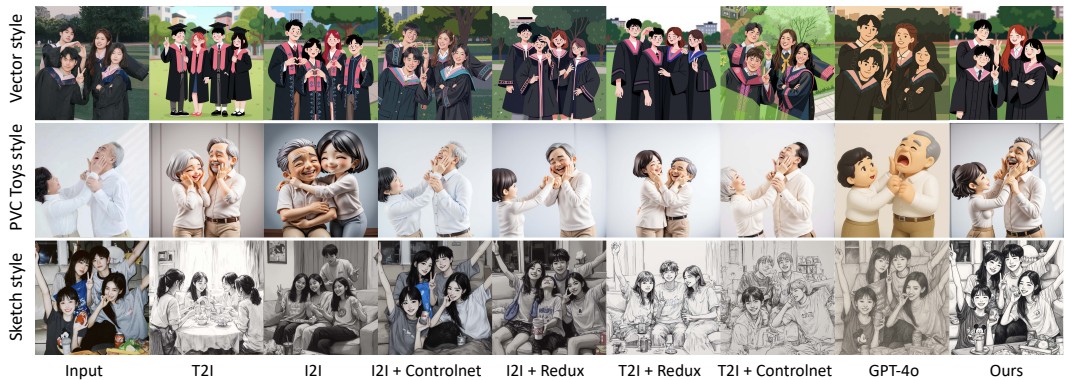

Figure 4: Comparation results of OmniConsistency and baseline methods.

ControlNet approach preserves structural alignment well, but introduces significant style degradation. In contrast, our method simultaneously achieves high style fidelity and content consistency, producing results comparable to the state-of-the-art GPT-4o.

Table 1: Grouped quantitative results on style, content, and text-image consistency.

| Method | Style Consistency | | | | | Content Consistency | | | Text-Img Align |
|---|---|---|---|---|---|---|---|---|---|
| | FID ↓ | CMMD ↓ | DreamSim ↓ | CLIP-I ↑ | GPT-4o ↑ | DreamSim ↓ | CLIP-I ↑ | GPT-4o ↑ | CLIP-S ↑ |
| Flux I2I | 44.4 | 0.168 | 0.236 | 0.783 | 4.38 | 0.307 | 0.704 | 4.27 | 0.277 |
| Flux I2I + Redux | 44.3 | 0.221 | 0.213 | 0.810 | 4.33 | 0.284 | **0.749** | 4.36 | 0.280 |
| Flux T2I + Redux | 39.4 | 0.186 | 0.218 | 0.871 | 4.49 | 0.320 | 0.707 | 4.40 | 0.316 |
| Flux I2I + CN | 70.0 | 0.736 | 0.265 | 0.761 | 4.14 | 0.315 | 0.742 | 4.48 | 0.290 |
| Flux T2I + CN | 60.2 | 0.556 | 0.247 | 0.801 | 4.37 | 0.322 | 0.738 | 4.44 | 0.297 |
| GPT-4o | - | - | - | - | - | 0.317 | 0.740 | **4.57** | 0.294 |
| **Ours** | **39.2** | **0.145** | **0.181** | **0.875** | **4.64** | **0.278** | 0.741 | 4.52 | **0.321** |

## 4.5 Ablation Study

**Ablation Study.** We conduct ablation experiments on two key design choices: (1) rolling training with multiple style LoRAs and (2) decoupled training of style and consistency. As shown in Fig. 5, when we remove rolling training and instead use a single LoRA trained on mixed-style data, the generated results maintain reasonable content consistency, but show a significant degradation in stylization quality on unseen styles. Moreover, when we remove the decoupled training strategy and directly train the consistency module together with style LoRA, both stylization capability and content consistency degrade notably, indicating strong entanglement between style and structure that harms overall performance.

Table 2: Ablation study with comprehensive metrics. Metrics are grouped by style consistency, content consistency, and text-image alignment.

| Variant | Style Consistency | | | | | Content Consistency | | | Text-Img Align |
|---|---|---|---|---|---|---|---|---|---|
| | FID ↓ | CMMD ↓ | DreamSim ↓ | CLIP-I ↑ | GPT-4o ↑ | DreamSim ↓ | CLIP-I ↑ | GPT-4o ↑ | CLIP-Score ↑ |
| Full Model (Ours) | **39.2** | **0.145** | **0.181** | **0.875** | **4.64** | **0.278** | 0.741 | **4.52** | **0.321** |
| w/o Rolling LoRA Bank | 47.5 | 0.266 | 24.98 | 0.849 | 4.14 | 0.322 | **0.762** | 4.48 | 0.319 |
| w/o Decoupled Training | 49.4 | 0.320 | 21.06 | 0.857 | 4.36 | 0.363 | 0.731 | 4.36 | 0.317 |

## 4.6 Discussion

We discuss the practicality and generality of OmniConsistency across three key aspects.

**Plug-and-Play Integration.** OmniConsistency is designed as a modular, plug-and-play component for maintaining consistency in image-to-image stylization. As shown in Fig. 6, it can be seamlessly combined with text-guided stylization, community LoRAs, or reference-based methods like IP-Adapter.

**Generalization to Unseen Styles.** Thanks to the decoupled training of style and consistency, along with the rolling LoRA Bank mechanism, OmniConsistency generalizes effectively to unseen style

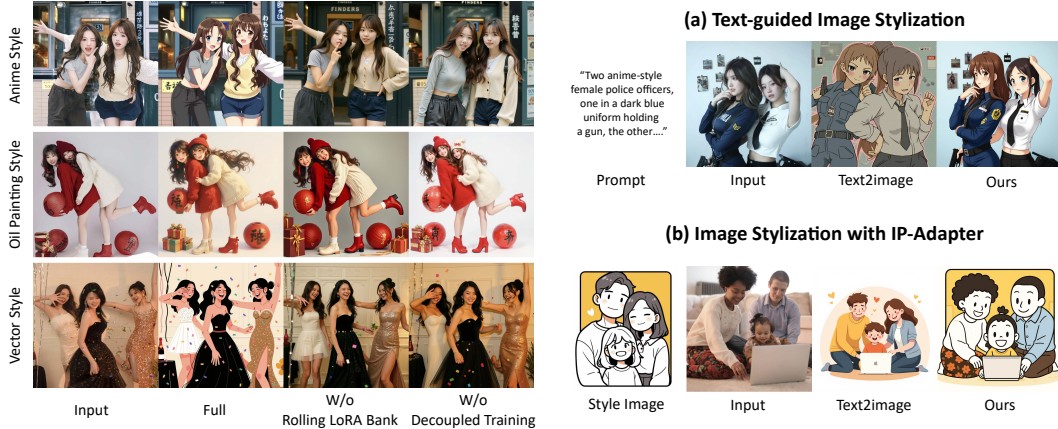

Figure 5: Ablation shows that full settings ensure strong stylization and consistency, while removals degrade performance.

Figure 6: OmniConsistency is plug-and-play and readily compatible with existing pipelines and tools like IP-Adapter.

Table 3: FID and CMMD scores across 10 styles (5 seen and 5 unseen).

| Metric | Seen Styles | | | | | | Unseen Styles | | | | | |
|---|---|---|---|---|---|---|---|---|---|---|---|---|
| | American Cartoon | Clay | Ghibli | Paper Cut | Van Gogh | Avg. | Comics | Oil Paint | Doll | Sketch | Vector | Avg. |
| FID ↓ | 37.6 | 37.9 | 42.2 | 36.4 | 31.3 | 37.08 | 41.3 | 41.8 | 35.9 | 39.9 | 37.0 | 39.18 |
| CMMD ↓ | 0.220 | 0.077 | 0.210 | 0.220 | 0.104 | 0.166 | 0.249 | 0.132 | 0.101 | 0.074 | 0.169 | 0.145 |

LoRA modules not seen during training. Fig. 3 shows qualitative examples, and Table 3 reports quantitative results (FID/CMMD) for both seen and unseen settings. Notably, there is no significant performance drop on unseen LoRAs compared to seen ones, indicating that OmniConsistency is style-agnostic and maintains strong generalization across diverse styles.

**High Efficiency.** Under the joint effect of several optimization strategies, OmniConsistency introduces only a marginal overhead compared to the base Flux Text-to-Image pipeline, incurring just a 4.6% increase in GPU memory usage and a 5.3% increase in inference time at 1024×1024 resolution with 24 sampling steps.

## 5 Limitation

We present several failure cases in the supplementary material. Specifically, our method has difficulty preserving non-English text due to limitations of the FLUX backbone, and may occasionally produce artifacts in small facial and hand regions.

## 6 Conclusion

houWe propose OmniConsistency, a plug-and-play consistency plugin for diffusion-based stylization that achieves full decoupling between style learning and consistency learning via a two-stage training strategy. Our method preserves identity, composition, and fine-grained details while generalizing well to unseen styles. It offers key advantages in plug-and-play compatibility, strong generalization, and high efficiency, making it suitable for integration with arbitrary LoRA styles without retraining. We also introduce a high-quality dataset across 22 diverse styles. Extensive evaluations demonstrate that OmniConsistency delivers state-of-the-art performance in both consistency and stylish quality, laying a solid foundation for controllable and high-fidelity image stylization.

## Acknowledgement

This project is supported by the National University of Singapore, under the Tier 1 FY2023 Reimagine Research Scheme (RRS).

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

# 7   Appendix

## 7.1   Implementation Details of the GPT-4o Evaluation

In the GPT-4o evaluation process, we establish specific metrics to assess various aspects of image generation tasks. These metrics are tailored to ensure comprehensive evaluation, capturing both objective scoring and comparative analysis for different types of tasks.

### 7.1.1   Direct Scoring Evaluation (for Style Transfer and Content Consistency Assessment)

The evaluation involves assessing the quality of the image generated through style transfer, considering both the consistency of the artistic style and the alignment with the original content. The scoring metrics used in this context include:

- **Style Consistency:** This measures how well the generated image reflects the artistic style of the reference images. The rating is provided on a scale from 1 (highly inconsistent) to 5 (extremely consistent).
- **Content Consistency:** This evaluates how closely the generated image mirrors the content of the original image, focusing on key elements such as facial features and overall layout. The scale ranges from 1 (highly inconsistent) to 5 (highly consistent).

For each aspect, the assistant provides a score based on a careful analysis of the image characteristics. The scores are then outputted in JSON format as follows:

```
{
"style_consistency": {
"score": 5,
"reason": "xxx"
},
"content_consistency": {
"score": 4,
"reason": "xxx"
}
}
```

### 7.1.2   Example of Task Prompt and Evaluation

Task Prompt: "Evaluate the style transfer of an image based on the provided reference style images and the original content image."

Images: [Upload images of the original content image, reference style images, and the generated images]

Evaluation: The assistant evaluates the generated image for both Style Consistency and Content Consistency, using the following criteria:

Style Consistency: How well does the generated image reflect the artistic style and overall atmosphere of the reference style images? The rating is given on a scale from 1 (highly inconsistent) to 5 (extremely consistent).

Content Consistency: How closely does the generated image resemble the content of the original image, including key elements like facial features and the overall layout? The rating is given on a scale from 1 (highly inconsistent) to 5 (extremely consistent).

This dual evaluation approach, focusing on both Style Consistency and Content Consistency, ensures a detailed and effective assessment of the quality of style transfer images generated by GPT-4o models.

## 7.2   User Study

### 7.2.1   Implementation Details

We conducted a user study through a questionnaire to evaluate the performance of different models in terms of style consistency and content consistency. A total of 30 questionnaires were distributed, each containing 30 questions. In terms of style consistency, we did not directly compare with GPT-4o because it does not support style LoRA injection. Instead, we approximated the desired style effects by carefully adjusting the prompts.

For each question, participants were provided with a reference image and the original image. They were then asked to select the best outputs for style consistency and content consistency from the results generated by different models (multiple selections allowed). During the analysis, each selection made for a particular model

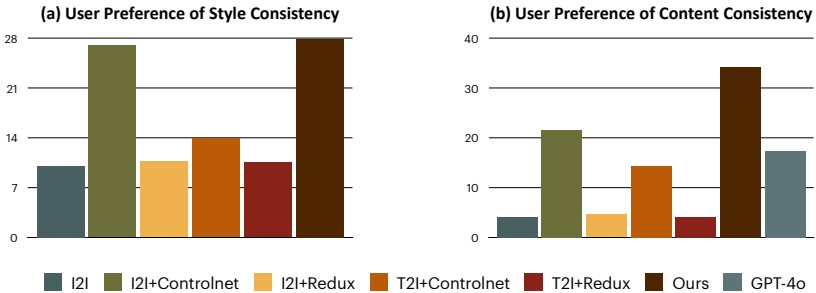

Figure 7: User study: Preference rates for style and content consistency across methods.

(a) Stylization of images containing non-English text  (b) Stylization of complex scenes with multiple people

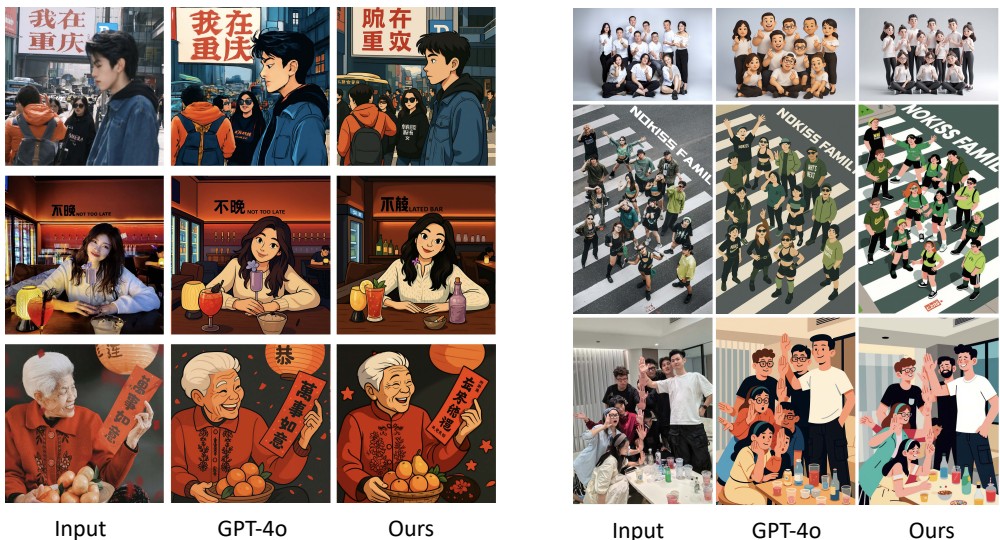

Figure 8: Failure cases.

was counted as one point, and the percentage score for each model was calculated based on the total number of selections. As shown in Fig. 7, our results received higher user preference in terms of both style consistency and content consistency.

### 7.2.2 Example of User Study

Question: Given the reference image and the original image, select the best outputs in terms of style consistency and content consistency from the provided options.

**Style Consistency:** How well does the generated image reflect the artistic style and overall atmosphere of the reference style images? Choose the best options from the provided images.

**Content Consistency:** How closely does the generated image resemble the content of the original image, including key elements such as facial features and overall layout? Choose the best options from the provided images.

### 7.3 Limitations and Failure Cases

We present several limitations and failure cases in Fig. 8. Specifically, Fig. 8 (a) illustrates stylization results on images containing Chinese text. While GPT-4o largely preserves the shape and legibility of the characters, our method struggles with maintaining the integrity of non-English text, likely due to limitations in the FLUX backbone. Fig. 8 (b) shows stylization outcomes on group photos and complex scenes. Both our method and GPT-4o occasionally exhibit inconsistencies in the number of people depicted, often omitting individuals who occupy smaller portions of the image. Additionally, artifacts may appear in small facial or hand regions.

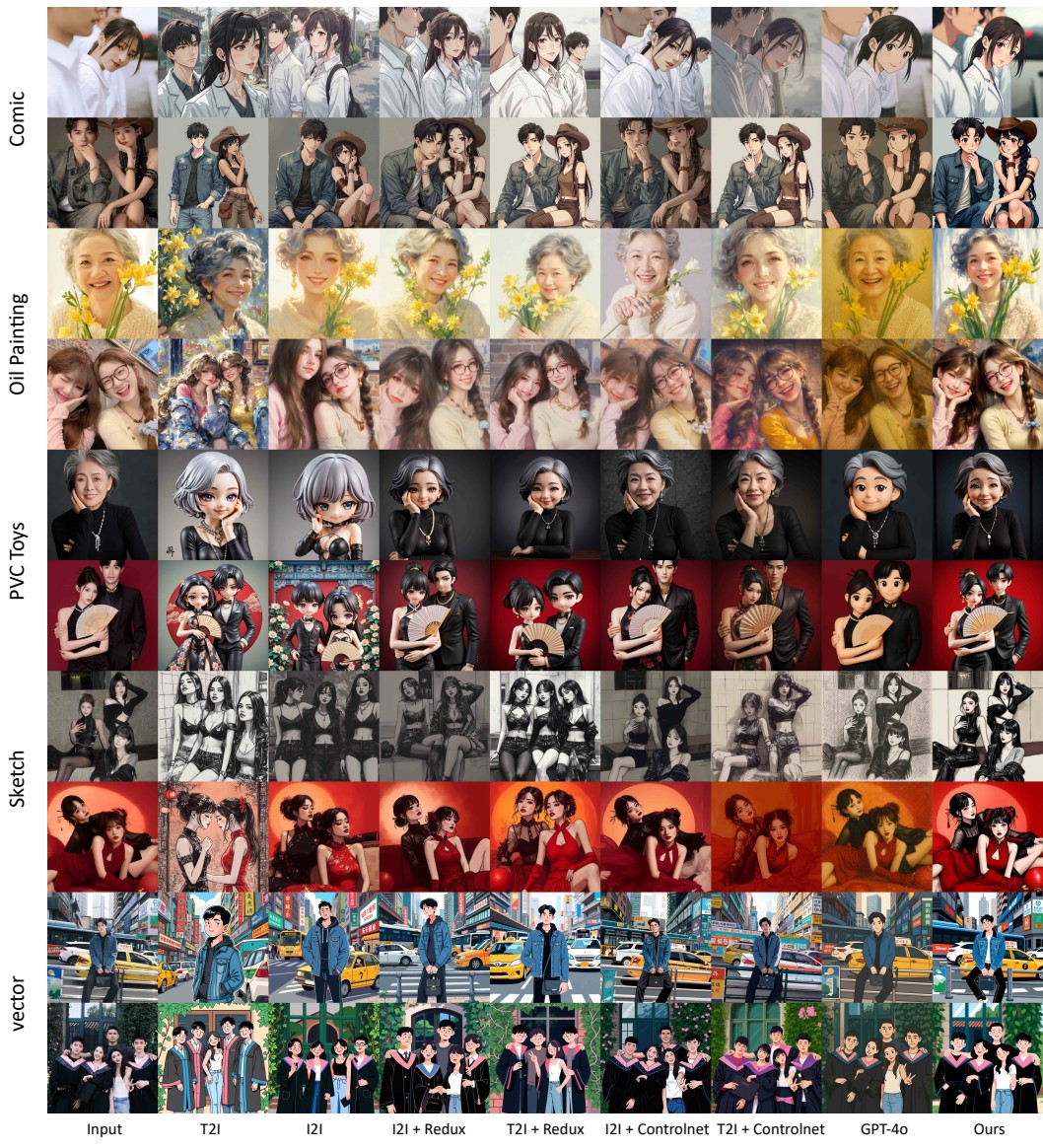

Figure 9: More Comparation results.

## 7.4 More Results

We present additional experimental results in this section. Fig. 9 shows the comparative results, while Fig. 10 and Fig. 11 demonstrates our method applied to a wider range of styles.

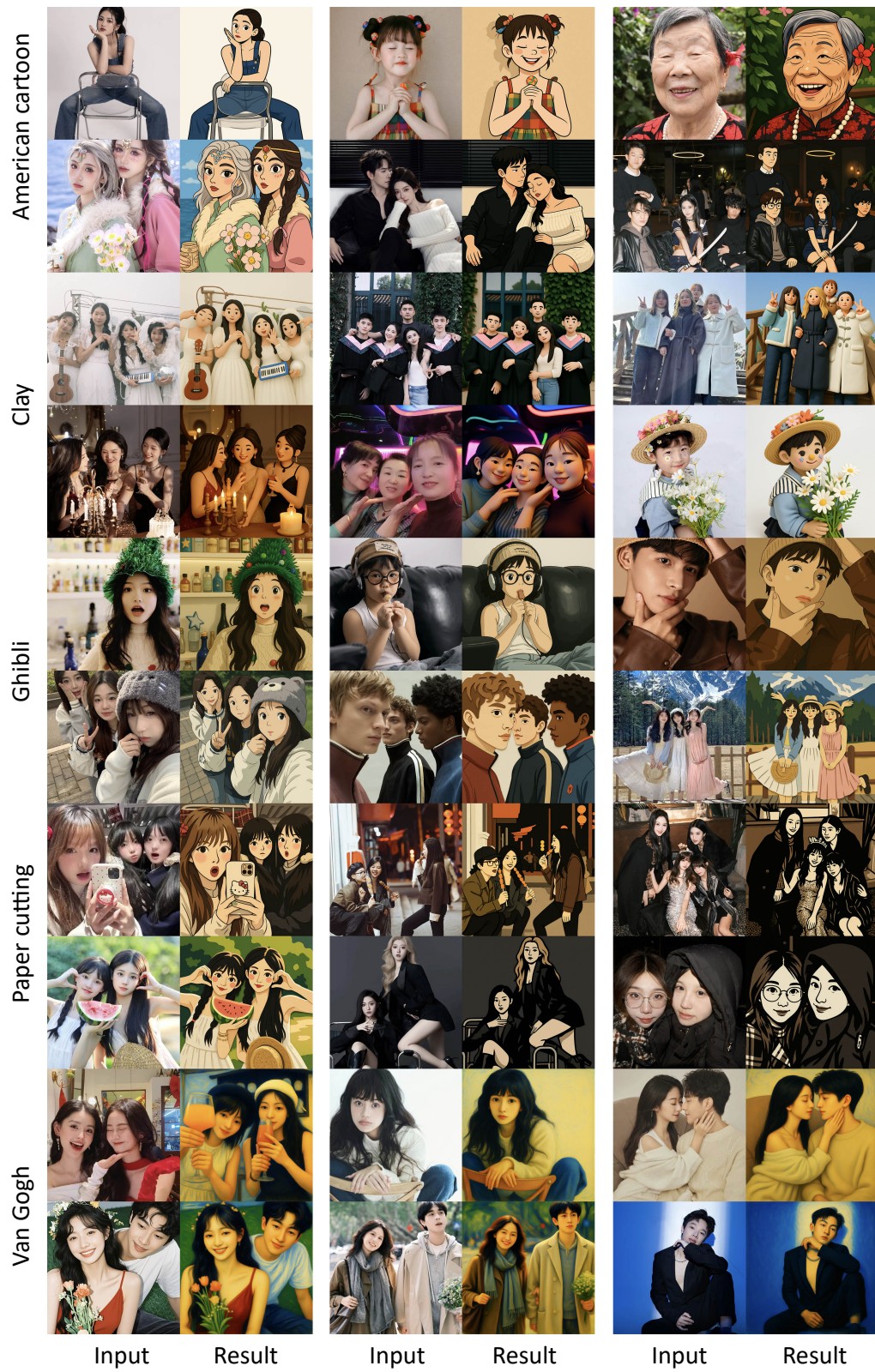

Input        Result        Input        Result        Input        Result

Figure 10: More image stylization results of OmniConsistency.

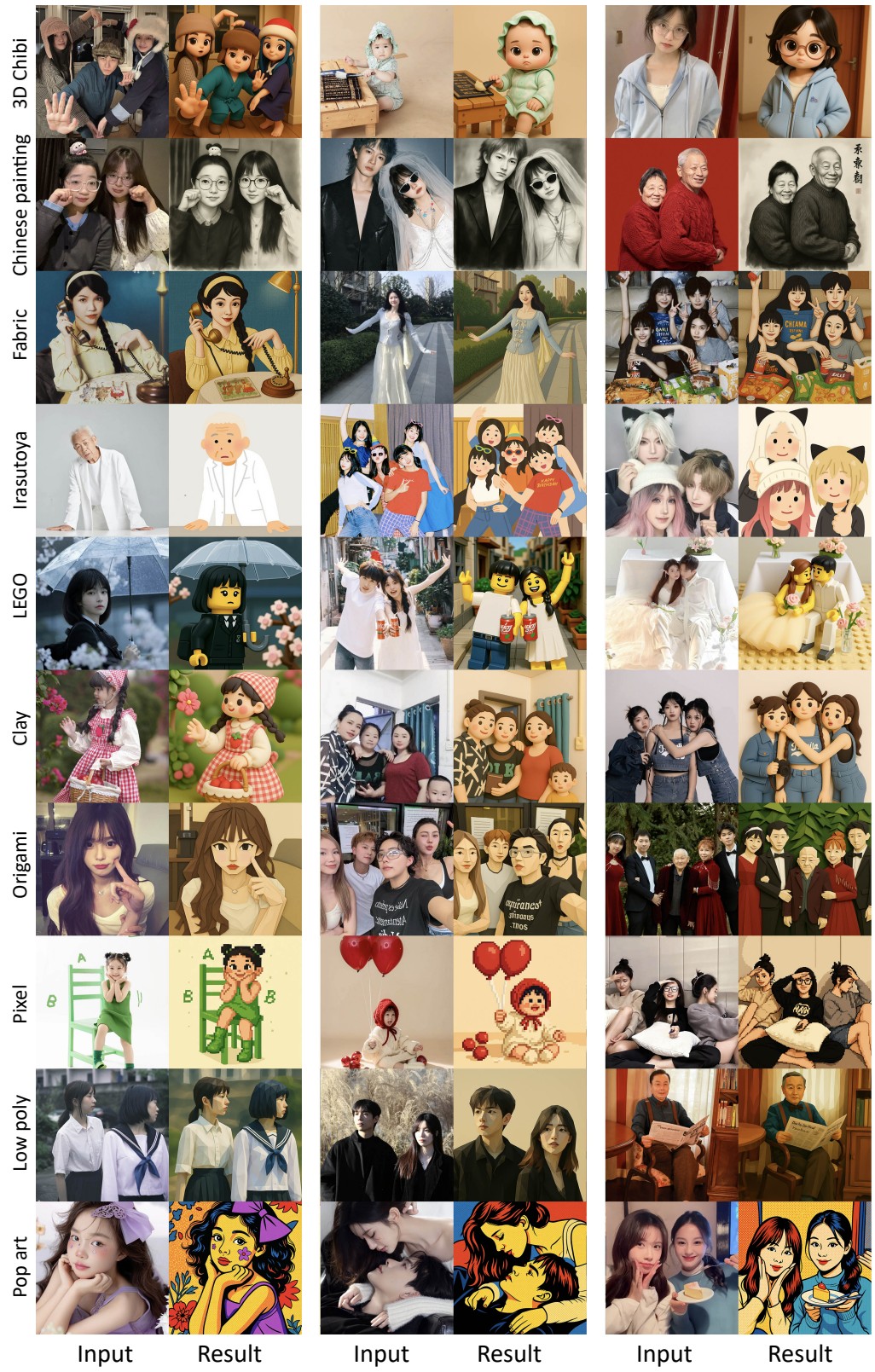

Figure 11: More image stylization results of OmniConsistency.

