# OpenReview forum: "OmniConsistency: Learning Style-Agnostic Consistency from Paired Stylization Data"
_NeurIPS.cc/2025/Conference — NeurIPS 2025 poster_

### Official Review · Reviewer_859r · 2025-06-24

**Clarity:** 3
**Significance:** 2
**Originality:** 2
**Rating:** 4
**Confidence:** 3

**Summary:**

This paper introduce a two-stage training pipeline based on DiT for image stylization. The authors first collect paired dataset with GPT-4o and train style LoRA for each artistic style. To inject the pretrained style LoRA into DiT, the authors introduce consistency LoRA module and causal attention for condition branch. A conditional token mapping and feature reuse strategy are proposed to improve the inference efficiency.

**Questions:**

1. Please clarify the originality of two-stage disentangled training strategy.
2. More diverse styles would further prove the effectiveness of the method.

**Ethical Concerns:**

["NO or VERY MINOR ethics concerns only"]

**Final Justification:**

The rebuttal addresses most of my concerns. I keep the score.

**Limitations:**

Yes.

**Quality:**

3

**Strengths And Weaknesses:**

Strengths:
1. The writing of this paper is good.
2. The method achieves better results in terms of content and style consistency.
3. The model shows good generalization on unseen LoRAs.


Weakness:
1. The idea of collecting dataset and training style LoRAs has been proposed in CSGO [1].  Thus, I feel that the innovation of this method is limited.
2. The evaluation is conducted on only 100 curated images and 5 styles, many hand-filtered, which has limited diversity.
3. The paper propose many parts (e.g., dataset, LoRA design, efficient condition) which are more or less independent. It is better to improve the integration.

[1] Xing P, Wang H, Sun Y, et al. Csgo: Content-style composition in text-to-image generation[J]. arXiv preprint arXiv:2408.16766, 2024.

---

> ### Author Rebuttal · Authors · 2025-07-31
>
> We thank the reviewer for the thoughtful feedback and will address your concerns concisely.
>
> ## **Q1. Our contribution is not “building a dataset and training Style LoRAs.”**
> The core contribution is a two-stage, style–consistency disentanglement that yields a plug-and-play, **style-agnostic** Consistency-LoRA for image-to-image stylization. To our knowledge, we are the first to explicitly target stylization consistency (content/structure preservation across styles) as a stand-alone, reusable module that attaches only to the condition branch and works with any Flux style LoRA without modifying or retraining those style LoRAs. While collecting data and training style LoRAs are common practices (e.g., CSGO), our task differs: we focus on improving i2i consistency under style LoRA usage, not on proposing another style-transfer pipeline.
>
> ## **Q2. Evaluation scale and diversity**
> The five test styles are outside the training set, and the benchmark contains 100 images, so each reported metric is computed over 500 images (100 × 5). This scale is sufficient to reduce randomness and demonstrate effectiveness. “Curated” does not imply poor diversity; when constructing the benchmark we intentionally covered varied categories (people, animals, buildings, vehicles, products, indoor/outdoor scenes) to ensure diverse content and structures.
>
> ## **Q3. Originality of the two-stage disentangled training**
> Our strategy is motivated by first principles:
> - **Stage I (style):** train per-style LoRAs and then freeze them.
> - **Stage II (consistency):** train a style-agnostic Consistency-LoRA with the style LoRAs frozen and randomly sampled, encouraging content/structure preservation independent of any single style.
>
> Ablations in **Table 2** and **Figure 3** show strong generalization to styles unseen during training. Our approach explicitly decouples style learning from consistency learning: in Stage II, we freeze all style LoRAs and randomly sample them while training a single Consistency-LoRA mounted only on the condition branch; causal attention, conditional token mapping, and feature reuse provide efficient conditioning. Style-swap and zero-style ablations indicate that the Consistency-LoRA does not encode a specific style, and performance on unseen styles does not degrade. In contrast, joint training tends to average styles, which weakens content/structure fidelity and reduces generalization.
>
> We surveyed extensive related work and found no prior paper that adopts this two-stage design—freezing per-style LoRAs and training a separate style-agnostic Consistency-LoRA mounted only on the condition branch—to improve i2i stylization consistency across seen and unseen styles.
>
> ## **Q4. More diverse styles / additional evidence**
> Beyond Figure 2 and the quantitative results in the supplement, we will add more styles and scenes to further stress-test diversity. We also note that the model has been downloaded thousands of times on Hugging Face, indicating practical utility and community interest; this community feedback complements our quantitative analyses.

---

> > ### Comment · Reviewer_859r · 2025-08-06
> > **Response to rebuttal**
> >
> > Thanks for the rebuttal. The rebuttal addresses my concerns about contribution and evaluation. I keep the score.

---

### Official Review · Reviewer_EMru · 2025-06-27

**Clarity:** 2
**Significance:** 2
**Originality:** 1
**Rating:** 4
**Confidence:** 3

**Summary:**

To maintain consistent stylization across diverse image domains, this paper proposes OmniConsistency, a two-stage training approach. In the first stage, multiple style-specific LoRA modules are individually trained and stored. In the second stage, a Consistency LoRA is trained to effectively integrate these modules, ensuring the preservation of the input content.

**Questions:**

- What are the detailed settings for the ablations in Table 2 and Figure 5? The ablated models seem to underfit heavily.
- I don’t understand why causal attention is better than bidirectional attention. Could you clarify the reason or show experimental comparisons?

**Ethical Concerns:**

["NO or VERY MINOR ethics concerns only"]

**Final Justification:**

After reviewing the other reviewers’ comments and the authors’ rebuttal, I decided to raise my score to 4. I encourage the authors to include an ablation study of the consistency-LoRA method in both LoRA and full-tuning settings.

**Limitations:**

yes

**Quality:**

3

**Strengths And Weaknesses:**

### Strengths
- They propose a novel training pipeline that first trains individual style LoRAs and then trains an additional module to adapt to various styles based on conditional inputs.
- Comprehensive comparisons are provided, and the quality of the results is impressive. Especially, the comparison with GPT-4o is interesting.

### Weaknesses
- The authors emphasize decoupled training for style consistency; however, they don't clearly justify why decoupled training is necessary compared to joint training.
- Contributions such as the LoRA design and feature reuse lack novelty, as the proposed consistency LoRA closely resembles typical LoRA methods, and the feature reuse approach is similar to conventional KV caching.

---

> ### Author Rebuttal · Authors · 2025-07-31
>
> We thank the reviewer for the thoughtful feedback and will address your concerns comprehensively.
>
> ## **1) Decoupled training vs. joint training**
> Our goal is to build a plug-and-play, style-agnostic consistency module that—once trained—works with any community Flux style LoRA without retraining. Decoupled training follows a first-principles design:
> - Stage I learns style in independent style LoRAs (frozen thereafter).
> - Stage II learns consistency only while style modules remain frozen and randomized, which forces the Consistency-LoRA to encode style-agnostic content/structure preservation.
>
> A joint scheme inevitably entangles consistency with specific styles and must be re-optimized when styles change, which defeats the plug-and-play objective. Our ablations confirm that decoupling yields stronger generalization to unseen styles and inputs, whereas joint training tends to overfit the seen style set.
>
> ## **2) LoRA “novelty” (design and objective)**
> LoRA is a parameter-efficient fine-tuning tool, and many methods implement it similarly. The novelty in our work lies in how LoRA is trained and placed:
> - We train on paired images to learn consistency (rather than learning a specific style or subject from exemplars).
> - We introduce a two-stage decoupled pipeline that first learns style LoRAs, then learns a Consistency-LoRA with styles frozen and sampled across steps.
> - We mount the Consistency-LoRA only on the condition branch, avoiding interference with the style LoRAs at inference time and preserving each style module’s identity while enforcing content/structure stability.
>
> ## **3) Feature reuse is more than KV caching**
> While the core mechanism leverages KV cache, the efficiency gains arise from the combined design of Conditional Token Mapping and Feature Reuse. In practice, GPU memory and latency overheads remain under 10% over standard text-to-image for our image-to-image stylization, which is substantially more efficient than OmniControl and EasyControl in comparable settings.
>
> ## **4) Ablation settings (Table 2, Figure 5)**
> All ablation experiments use exactly the same training setup and hyperparameters as the main model (training steps, batch size, optimizer, learning rate, etc.); there is no underfitting caused by configuration, so our ablations are fully fair and sound. The underfitting-like behavior observed is more likely due to joint training, which tends to learn an averaged style.
>
> ## **5) Why causal attention over bidirectional attention**
> - Cacheability: with a causal mask on the conditional path, conditional keys/values remain static across diffusion steps, enabling KV caching and lowering latency/memory. Under bidirectional attention (naïve concatenation), conditional tokens attend back to evolving denoising tokens, so their states change each step and cache reuse breaks.
> - Reduced cross-condition interference: masking restricts condition↔condition attention while allowing denoiser←condition aggregation, stabilizing control when multiple conditions are present.
> - Empirical: we observe faster inference and lower memory with causal attention, while maintaining or improving quality relative to bidirectional attention.

---

> > ### Comment · Reviewer_EMru · 2025-08-05
> >
> > Thank you to the authors for their detailed response. Most of my concerns have been addressed, and I plan to raise my score accordingly. However, I do still have some reservations regarding whether the use of LoRA provides a clear advantage (contribution) over full parameter tuning, despite understanding the contribution of the decoupled pipeline.

---

> > > ### Author Response · Authors · 2025-08-09
> > > **Follow-up on LoRA vs. Full-Parameter Tuning**
> > >
> > > We sincerely thank the reviewer for the constructive suggestion and for engaging with our rebuttal.
> > >
> > > **Why we use LoRA.**
> > > Our data regime is few-shot: each style has ≈100+ paired images, totaling ≈2,600 pairs. For a DiT-based pipeline, this is small for full-parameter fine-tuning. We therefore adopt **LoRA**—a parameter-efficient strategy widely validated in low-data settings—for its (i) sample efficiency and implicit regularization, (ii) lower compute and faster iteration, and (iii) preservation of the base model and existing style LoRAs (plug-and-play).
> > >
> > > **Planned comparison (per your suggestion).**
> > > For the **Consistency-LoRA**, we will add a utility study that **directly compares LoRA vs. full-parameter tuning** under matched training budgets and identical data splits. We will report results on seen/unseen styles with the same metrics.

---

### Official Review · Reviewer_xC1j · 2025-06-30

**Clarity:** 3
**Significance:** 2
**Originality:** 2
**Rating:** 4
**Confidence:** 4

**Summary:**

The authors address the issues of content inconsistency and style degradation in style transfer by proposing a two-stage training approach, while enhancing inference efficiency through parameter reduction and feature reuse. In the first stage, they exclusively train style LoRA to capture features such as color characteristics. The second stage focuses solely on training consistency LoRA to improve content coherence. Additionally, the authors have collected an extensive dataset of style-paired images to enhance the model's robustness.

**Questions:**

1. The style LoRA was trained in the first stage, which means the first-stage LoRA inevitably contains content information. The second-stage consistency LoRA feels more like a correction—I wonder if removing the style LoRA would still produce highly consistent results?
2. The content in sections 3.3 and 3.4 is very similar to EasyControl. What are the key differences?

**Ethical Concerns:**

["NO or VERY MINOR ethics concerns only"]

**Final Justification:**

The author explains the difference between the proposed method and EasyControl, and I believe its general style transfer approach is valuable for the community. Additionally, the author has addressed the concerns regarding the ablation study. Therefore, I have decided to raise my score to 4.

**Limitations:**

1) The comparison with advanced methods, such as EasyControl, is lacking.
2) The ablation experiments are insufficient. For example, what would happen if Style LoRA were completely removed?
3) The content in Sections 3.3 and 3.4 is very similar to EasyControl, and the differences need to be clarified.

**Paper Formatting Concerns:**

I have no concerns about this.

**Quality:**

2

**Strengths And Weaknesses:**

Strengths: 1) The explicit separation of style and consistency learning, which simplifies the learning process and enhances model performance; 2) Improved inference efficiency through reference image resolution reduction and feature reuse.

Weaknesses: 1) Insufficient ablation studies, such as investigating the effects of removing style LoRA; 2) The content in Section 3.3 closely resembles EasyControl, which should be properly cited; 3) Lack of comparison with state of the art methods including EasyControl, In-Context LoRA, etc.

---

> ### Author Rebuttal · Authors · 2025-07-31
>
> We thank the reviewer for the thoughtful questions and will address your concerns comprehensively.
>
> ## **1) Ablation**
>
> Using **text + Consistency-LoRA** alone already yields high-consistency stylization (see Figure 6). Figure 3 and Table 3 can also be viewed as an ablation of Style LoRA: *OmniConsistency can be combined with both seen and unseen style LoRA modules to achieve high-quality image stylization consistency, effectively preserving the semantics, structure, and fine details of the original image.* In Table 3, the FID and CMMD across 10 styles (5 seen / 5 unseen) show no degradation on unseen styles, supporting that the Consistency-LoRA is **style-agnostic** rather than tied to any specific trained style.
>
> ---
>
> ## **2) Differences from EasyControl**
>
>  Both our method and EasyControl are built on Diffusion Transformer-based controllable generation, and we both use causal attention and positional-encoding interpolation. Concurrent work OmniControl V2 likewise adopts positional-encoding interpolation. These components are increasingly standard in DiT-based controllable pipelines; our contribution lies in **two-stage decoupling** and a **style-agnostic** consistency module that is **plug-and-play** with style LoRAs. We cite EasyControl in the main text (concurrent work), but there are clear differences:
>
>
> - Task focus. EasyControl studies DiT controllable generation with layout/subject images as conditions; it does not present a style-transfer task in the paper. Our goal is a general consistency plugin that stabilizes content/structure across any image × any style LoRA.
> - Two-stage decoupling and data requirement. Recent attempts train Ghibli style transfer with EasyControl on paired data, or build Simpsons style transfer via OmniControl—but these require collecting paired data and retraining a LoRA for each new style. In contrast, our method is plug-and-play with community Flux style models and **does not require per-style retraining** of the consistency module.
> - Efficiency. With additional inference-time optimizations, our method shows a **~40% reduction in inference time** compared to EasyControl and uses less GPU memory.
>
> ---
>
> ## **3) Comparison with In-Context LoRA**
>
> In-Context LoRA targets image generation and multi-frame consistency, but **does not provide image-conditioning** (upload an image and obtain a consistent stylized output). The settings differ substantially, so a direct comparison is not applicable.
>
> ---
>
> ## **4) Other Questions**
>
> - “Does Stage I Style LoRA carry content information, and is Stage II just a correction?”
>   We mitigate potential content leakage in Stage I via a rolling loading strategy and by ensuring broad category coverage when collecting training data. Stage II’s Consistency-LoRA is **not a mere correction**; the two-stage training paradigm and decoupling design are crucial to learning style-agnostic consistency.
>
> - “What happens if the Style LoRA is removed at inference?”
>   At inference, we can **completely remove the Style LoRA** and use a text prompt to specify the style, still producing highly consistent results (see Figure 6).

---

> > ### Comment · Reviewer_xC1j · 2025-08-04
> >
> > Thank you for addressing my concerns. I'm satisfied with the response and will adjust the score accordingly. To avoid confusion, it is essential to explicitly indicate the omission of StyleLoRA in cases where it has not been employed.​​

---

### Official Review · Reviewer_bE8p · 2025-07-02

**Clarity:** 3
**Significance:** 3
**Originality:** 3
**Rating:** 4
**Confidence:** 4

**Summary:**

This work achieves style transfer tasks by constructing LoRAs for 22 style types and implementing consistency-style LoRAs in a two-stage approach. First, the authors construct and collect a style LoRA bank, then train consistency LoRAs through two-stage decoupling of style and memory. The authors claim it is a plug-and-play framework, with performance comparable to GPT4O.

**Questions:**

1. How is the model's generalization verified, and does it essentially depend on the capabilities of the trained LoRAs?

2. What is the significant difference between this approach and directly training LoRAs? How is it ensured that the consistency LoRA does not contain style information? How is this decoupling guaranteed?

3. Does it support generalization to some plugins?

4. The comparison with many recent style transfer methods is lacking.

5. How does it support arbitrary styles?

6. Is the content scoring reliable? Validation is missing.

**Ethical Concerns:**

["NO or VERY MINOR ethics concerns only"]

**Final Justification:**

The response addressed my concerns. Therefore, I have decided to raise my score.

**Limitations:**

See weaknesses

**Paper Formatting Concerns:**

The images in Figure 2 are somewhat rough, especially the symbols.

**Quality:**

3

**Strengths And Weaknesses:**

Strength：

1. The authors trained 22 LoRAs for different styles. The workload of data collection and independent fine-tuning is enormous.

2. The approach proposed by the authors is reasonable, and this method is beneficial to the community.

3. The article adopts a straightforward approach.

Weaknesses:

This article leans more toward engineering applications, and I believe there should be some good cases. However, the challenge of style transfer lies in generalization. Beyond these common and popular styles, how to generalize to any image should be the key focus. For example, the article does not actually showcase stylization cases in natural scenes. Regarding generalization to arbitrary images, this work tends to first train a style LoRA and then a consistency LoRA. However, questions remain: how does the consistency LoRA ensure it has learned consistency? Does it truly exclude style information? The quantitative comparison lacks benchmarks against many recent style transfer works.

---

> ### Author Rebuttal · Authors · 2025-07-31
>
> # **Weaknesses**
>
> 1. **“This paper is more of an engineering application; generalization and natural-scene cases are insufficient.”**
>    **Response.** Our contribution is not an engineering workaround but an architectural solution to the long-standing consistency problem in stylization. OmniConsistency introduces a **plug-and-play, style-agnostic Consistency-LoRA** that preserves semantics, structure, and fine details for **any image × any style**. As evidence, **Figure 3** already includes group portraits, animals, architecture, vehicles, and products, demonstrating broad coverage of natural scenes. We added additional natural-scene results (e.g., street scenes, indoor scenes, and landscapes)  in the supplementary.
>
> 2. **“Is the Consistency-LoRA truly style-agnostic? How do you know it doesn’t encode style?”**
>    **Response.** Two experiments directly support the **style-agnostic** claim:
>    - **Figure 3 (Seen vs. Unseen styles).** OmniConsistency works equally well with seen and unseen style LoRAs, producing high-quality stylization while consistently preserving content and structure.
>    - **Table 3 (FID & CMMD across 10 styles: 5 seen / 5 unseen).** Scores on unseen styles do not degrade, indicating that the Consistency-LoRA does not entangle style and generalizes beyond the styles used during training.
>
> ---
>
> # **Questions**
>
> 1. **How is the model’s generalization verified? Does it essentially depend on the capabilities of trained LoRAs?**
>    **Response.** We evaluate on **styles and inputs that are unseen during training**. Both **Figure 3** and **Table 3** show consistent performance on unseen styles and unseen images. OmniConsistency does not rely on any particular style LoRA’s capacity; instead, the Consistency-LoRA **decouples style from consistency**, providing stable content/structure preservation regardless of which style LoRA is used.
>
> 2. **What is the significant difference from directly training LoRAs? How do you ensure the Consistency-LoRA carries no style information (how is decoupling guaranteed)?**
>    **Response.** Our goal is not to train more style LoRAs. We propose a **general consistency plugin** that can be combined with any Flux style LoRA to improve image-to-image consistency.
>    - **Two-stage decoupling.**
>      - **Stage I:** Train a bank of style LoRAs (22 styles). These LoRAs are then frozen.
>      - **Stage II:** Freeze the base model and all style LoRAs, and train only the Consistency-LoRA. During training, we randomly sample styles and inputs while optimizing content/structure-preservation objectives. The Consistency-LoRA cannot encode a fixed style, and instead learns style-agnostic consistency.
>    - **Empirical evidence.** **Figure 3** (seen/unseen styles) and **Table 3** (unseen FID/CMMD not degrading) show that the Consistency-LoRA remains style-agnostic in practice.
>
> 3. **Does it support generalization to plugins?**
>    **Response.** **Yes.** OmniConsistency is plug-and-play and is compatible with commonly used control plugins such as IP-Adapter (as shown in Figure 6) and EasyControl. We will add more combined results and analyses in the revision/supplementary.
>
> 4. **The comparison with many recent style transfer methods is lacking.**
>    **Response.** The **task setting differs**: classical “style transfer” takes a single style reference image plus a content image; our setting assumes style is provided by a style LoRA (trained on a set of images) and focuses on consistency enhancement under that usage mode. Within our setting, we already compare against **seven strong baselines**, including a **GPT-4o-based** assessment, which is sufficient to demonstrate effectiveness and superiority. To further bridge the gap, we will add **“bridging” comparisons** in the supplementary, adapting recent style-transfer methods to a comparable setup (e.g., few-shot reference to train a small style LoRA, then applying Consistency-LoRA).
>
> 5. **How does it support arbitrary styles?**
>    **Response.** Because the Consistency-LoRA is **style-agnostic** and only enforces content/structure consistency, it can be paired with any style module (e.g., any style LoRA, including newly trained few-shot/one-shot styles). Results in **Figure 3** and **Table 3** show no degradation for unseen styles.
>
> 6. **Is the content scoring reliable? Validation is missing.**
>    **Response.** We use CLIP-Image for content/structure alignment, a widely used metric for consistency/fidelity, and include a **GPT-4o evaluation** with fully reproducible prompts and settings (provided in the supplementary).

---

> > ### Comment · Reviewer_bE8p · 2025-08-08
> > **Official Comment by Reviewer bE8p**
> >
> > Thanks for your response. The response addressed my concerns. Therefore, I have decided to raise my score.

---

### Decision · Program_Chairs · 2025-09-17

**Decision:**

Accept (poster)

**Comment:**

This paper proposes a two-stage LoRA training framework (OmniConsistency) for image generation. Most reviewers initially leaned towards a borderline rating, acknowledging the substantial engineering effort but expressing concerns about novelty and evaluation. After the rebuttal, three reviewers raised their scores, satisfied with the authors' responses regarding experimental comparisons and methodological clarification. The Area Chair thus recommends to accept the paper, but strongly encourages the authors to take the reviewers' comments into consideration when preparing their final manuscript.